# Clinical Outcomes and Adverse Effects in Septic Patients with Impaired Renal Function Who Received Different Dosages of Cefoperazone–Sulbactam

**DOI:** 10.3390/antibiotics11040460

**Published:** 2022-03-29

**Authors:** Chien-Hsiang Tai, Hung-Jen Tang, Chen-Hsiang Lee

**Affiliations:** 1Division of Infectious Diseases, Department of Internal Medicine, Kaohsiung Chang Gung Memorial Hospital, Kaohsiung 833, Taiwan; idchtai@gmail.com; 2Department of Medicine, Chi Mei Medical Center, Tainan 710, Taiwan; 8409d1@gmail.com; 3Department of Medical Research, Chi Mei Medical Center, Tainan 710, Taiwan; 4Department of Health and Nutrition, Chia Nan University of Pharmacy and Sciences, Tainan 717, Taiwan; 5College of Medicine, Chang Gung University, Kaohsiung 833, Taiwan

**Keywords:** multidrug-resistant organism, ß-lactam and ß-lactamase inhibitors, renal insufficiency, prothrombin time, coagulopathy

## Abstract

This study aimed to compare clinical outcomes and adverse effects in septic patients with impaired renal function who received different dosages of cefoperazone–sulbactam (CFP–SUL 1 g/1 g or 2 g/2 g every 12 h). The retrospective study was conducted using the Chang Gung Research Database to include adult patients who had renal insufficiency presented with septicemia caused by Gram-negative organisms and had received CFP–SUL for more than 1 week. A total of 265 patients (44 in the CFP–SUL 1 g/1 g group and 221 in the CFP–SUL 2 g/2 g group) were eligible to be included in this study. After 1:3 propensity score matching, 41 and 123 patients in the CFP–SUL 1 g/1 g and CFP–SUL 2 g/2 g groups, respectively, were included for analyses. There were no significant between-group differences in all-cause mortality rates and adverse effects, including prolonged prothrombin time. A logistic regression model showed that the Pitt bacteremia score was related to all-cause mortality rate and prolonged prothrombin time was associated with renal replacement therapy. The adverse effects of CFP–SUL did not increase in septic patients with impaired renal function receiving CFP–SUL 2 g/2 g Q12H. However, this study may be underpowered to reveal a difference in all-cause mortality.

## 1. Introduction

The Centers for Disease Control and Prevention advocates that several Gram-negative bacteria, such as *Acinetobacter baumannii*, *Pseudomonas aeruginosa*, and extended-spectrum beta-lactamase (ESBL)-producing *Enterobacteriaceae*, need public attention because of emerging threats of antimicrobial resistance [1,2]. A five-year observational study showed that major pathogens causing lower respiratory tract infections were *Acinetobacter baumannii*, *Staphylococcus aureus*, *Pseudomonas aeruginosa*, and *Klebsiella pneumoniae*, and most of the isolates were resistant to cephalosporins and carbapenems [3].

These multidrug-resistant organisms (MDROs) manifest concomitant resistance to commonly used classes of antimicrobials [4]. Although there are some new drugs, such as meropenem/vaborbactam or cefiderocol, available now, high medical cost might interfere with the wide availability of these new antibiotics [5]. This has forced clinicians to consider treatment approaches based on combinations of drugs with impaired activity and/or to rediscover old drugs with suboptimal pharmacokinetics and toxicity issues [6].

Cefoperazone–sulbactam (CFP–SUL, Brosym^®^; TTY Biopharm Company, Taipei, Taiwan) is a combination of ß-lactam and ß-lactamase inhibitors. CFP–SUL has been proven to be effective in treating hospital-acquired pneumonia, ventilator-associated pneumonia, and bloodstream infection [7,8,9]. They have also been used to treat infections caused by MDROs, including ESBL-producing *Enterobacteriaceae* [10,11,12], because sulbactam can enhance the activity of cefoperazone against *A. baumannii* [13,14]. With emerging resistance to the third generation of cephalosporins in *Enterobacteriaceae* in bloodstream infections, CFP–SUL may be useful in empirically treating bloodstream infections [15]. In addition, some antibiotics, such as cefoxitin and clavulanic acid, can induce AmpC β-lactamase expression [16,17]; however, sulbactam does not induce class I (AmpC) chromosomal β-lactamases in *Enterobacteriaceae* [14].

Cefoperazone can be administered without the need for adjustment in renal dysfunction [18]. As for sulbactam, a meta-analysis suggested that a higher dose of sulbactam (≥6 g/day) combined with either levofloxacin, minocycline, or tigecycline could achieve better clinical efficacy in treating multidrug-resistant *A. baumannii* (MDR-AB) in patients with normal renal function [19]. Regarding patients with poor renal function, there are limited data on the optimal dosage of sulbactam, although the dosage of ampicillin–sulbactam needs to be adjusted if creatinine clearance is less than 50 mL/min/1.73 m^2^ [20]. One pharmacokinetic–pharmacodynamic study suggested a higher dosage of sulbactam (2 g every 6 h) for patients with renal insufficiency in the treatment of MDR-AB infection [21]. According to the package insert, the dosage of CFP–SUL needs to be adjusted from 2 g/2 g to 1 g/1 g every 12 h (Q12H) if creatinine clearance is below 30 mL/min/1.73 m^2^. In treating infections caused by MDROs, the need for dosage adjustment of CFP–SUL in patients with renal insufficiency remains unknown. Furthermore, the most concerning adverse effect of CFP–SUL is prolonged prothrombin time (PT), yet little is known about whether it increases the risk of prolonged PT when no adjustment of CFP–SUL dosage is made in patients with impaired renal function [22]. Therefore, the goal of our study was to investigate differences in clinical outcomes and side effects between administration of CFP–SUL 1 g/1 g and 2 g/2 g Q12H in patients with septicemia caused by Gram-negative microorganisms who had impaired renal function.

## 2. Results

### 2.1. Clinical Characteristics

A total of 7469 patients who had been administered with CFP–SUL were identified from the Chang Gung Research Database (Figure 1). After excluding patients infected by Gram-positive organisms or fungi (*n* = 4113), those who had received CFP–SUL within 1 week before a septicemic episode (*n* = 1073), those with estimated glomerular filtration rate (eGFR) >60 mL/min/1.73 m^2^ or receiving continuous renal replacement therapy (*n* = 214), those receiving CFP–SUL for <7 days (*n* = 1799) and those receiving concomitant warfarin or direct oral anticoagulant (*n* = 5) were included, making up a total of 265 patients. Among these, 44 patients were in the CFP–SUL 1 g/1 g group and 221 in the CFP–SUL 2 g/2 g group. In total, 47.7% of the patients were men. Their mean age was 74.9 ± 13.0 years, Charlson comorbidity index (CCI) was 2.6 ± 3.2, Pitt bacteremia score (PBS) was 2.1 ± 2.5, and body weight was 54.4 ± 6.6 kg. After 1:3 propensity score matching (PSM) by sex, age, CCI, and PBS, 41 and 123 patients in the CFP–SUL 1 g/1 g and 2 g/2 g groups, respectively, were included for further analyses. Table 1 shows the clinical characteristics before and after PSM. Although there was a significant difference in the proportions of congestive heart failure and mild liver disease between the two groups, the difference disappeared after PSM (Table 1).

### 2.2. Clinical Outcomes and Adverse Effects

As shown in Table 2, there was no significant difference in 14-day and 30-day all-cause mortality in the unmatched and PSM cohorts; however, there was a trend of higher mortality rate in the CFP–SUL 1 g/1 g group. The CFP–SUL 2 g/2 g group seemed to have a better clinical resolution rate, although the difference did not reach statistical significance. Within 7 days after the end of treatment, 90 patients had repeated blood cultures. None of the 12 patients in the CFP–SUL 1 g/1 g group and 1 out of 78 patients (1.3%) in the 2 g/2 g group had the same pathogen isolated as the previous septicemic episode. We found that 38.6% and 37.6% of patients had abnormal international normalized ratio (INR) in the CFP–SUL 1 g/1 g and 2 g/2 g groups, respectively, before PSM (*p* = 0.89). Likewise, in the PSM cohort, 36.6% of patients in the CFP–SUL 1 g/1 g group and 40.7% in the CFP–SUL 2 g/2 g group developed coagulopathy (*p* = 0.64). There were no statistically significant differences between the two groups in terms of adverse effects, such as *Clostridioides difficile*-associated diarrhea (CDAD), diarrhea, leukopenia, and neutropenia. Figure 2 shows the risk factors for abnormal INR. Compared to those with eGFR of 30–60 mL/min/1.73 m^2^, patients who received renal replacement therapy were more likely to have coagulopathy (odds ratio, 2.24; 95% CI, 1.17–4.29). Dosage of CFP–SUL 1 g/1 g or 2 g/2 g Q12H, on the other hand, was not related to coagulopathy.

As shown in Table 3a, there were no statistically significant differences in 14-day and 30-day all-cause mortality rates, clinical resolution, or adverse effects, including PT prolongation, CDAD, diarrhea, leukopenia, and neutropenia, between the two groups with bacteremia due to MDROs, although the mortality rate seemed to be higher in the CFP–SUL 2 g/2 g group without reaching statistical significance. However, the proportion of bacteremia caused by *Acinetobacter* spp. was significantly higher in the CFP–SUL 2 g/2 g group than in the CFP–SUL 1 g/1 g group (39.2% vs. 4.3%, *p* < 0.01, Table 3b).

Table 4 indicates that 14-day and 30-day all-cause mortality were unrelated to the dosage of CFP–SUL administered in both univariate and multivariate analyses. PBS with a cut-off value of 4 was found to be an independent factor associated with mortality. There was no significant evidence of lack of fit in any of the final models as the *p* values were >0.05 in the Hosmer–Lemeshow goodness-of-fit tests.

## 3. Discussion

The current study indicated that there was no significant difference in the incidence of adverse effects, including abnormal INR, among patients with impaired renal function who received CFP–SUL 2 g/2 g Q12H and those who received 1 g/1 g Q12H for treatment of septicemia caused by Gram-negative organisms. Regarding clinical outcomes of patients with septicemia caused by Gram-negative organisms (Table 2), the 14-day and 30-day all-cause mortality seemed to be higher in the CFP–SUL 1 g/1 g group than in the CFP–SUL 2 g/2 g group. The rate of clinical resolution also tended to be higher in the CFP–SUL 2 g/2 g group than in the CFP–SUL 1 g/1 g group. Cefoperazone can be administered without adjustment of dosage for renal impairment [18,23]. Previous trials have demonstrated the efficacy of cefoperazone in the treatment of sepsis. The dosage of cefoperazone in most trials was at least 2 g, twice a day [24]. There are scarce data on the clinical efficacy of an adjusted dosage of cefoperazone for renal insufficiency [18]. Therefore, in our study, the better outcomes of patients in CFP–SUL 2 g/2 g group compared to patients in the CFP–SUL 1 g/1 g group regarding mortality rate and clinical resolution rate are probably due to the suboptimal dosage of cefoperazone [24]. However, the mortality and clinical resolution rates did not reach statistical significance, probably due to the relatively small sample size of the study or the fact that some confounding factors needed to be adjusted, such as the difference in pathogens that caused septicemia (Table 2).

A major safety concern of CFP–SUL is coagulation disorder, mainly PT prolongation, by interfering with the metabolism of vitamin K, which can sometimes be fatal [22,25]. Our study illustrated that the incidence of adverse effects did not differ between the two groups. The incidence of abnormal INR was not higher in patients receiving CFP–SUL 2 g/2 g Q12H. Moreover, the incidence of CDAD did not differ between the two groups.

As for bacteremia caused by MDRO, the mortality rate, although not significantly different, seemed to be higher in the CFP–SUL 2 g/2 g group. However, the MDROs causing bacteremia differed between the two groups. In the CFP–SUL 1 g/1 g group, only one patient had bacteremia due to *Acinetobacter* spp. (4.3%). On the other hand, in the CFP–SUL 2 g/2 g group, 40 patients (39.2%) had septicemia caused by *Acinetobacter* spp. (*p* < 0.01). *Acinetobacter* spp. has been reported to be an independent risk factor for lower survival rates for bacteremia [26,27]. Sulbactam plays a major role in the treatment of infections caused by *A. baumannii*, [19,28]. In vitro studies have also shown that sulbactam could enhance the activity of cefoperazone against *A. baumannii* [29], especially with a higher proportion of sulbactam in CFP–SUL [13]. In this study, most patients with septicemia caused by ceftriaxone-resistant *Enterobacteriaceae* received CFP–SUL 1 g/1 g Q12H treatment. In addition, one patient with septicemia caused by *Acinetobacter* spp. was treated with CFP–SUL 1 g/1 g Q12H. This might be a potential bias between the different dosages received by the CFP–SUL groups regarding mortality in bacteremia caused by MDROs. In addition, the severity of patients with septicemia caused by MDROs could not be stratified by PSM because of the small sample size.

The logistic regression model revealed that both 14-day and 30-day all-cause mortality rates were not related to the dosage of CFP–SUL administered. In contrast, PBS was an independent risk factor for mortality. Our results are consistent with those of a previous study showing that PBS is a risk factor for mortality in patients with Gram-negative bloodstream infections [30]. Our study also found that CFP–SUL 2 g/2 g Q12H was not a risk factor for abnormal INR; instead, patients with renal replacement therapy had a higher risk of abnormal INR than those with eGFR of 30–60 mL/min/1.73 m^2^ (Figure 2). Cefoperazone-related products cause coagulopathy through vitamin-K-dependent thrombin factor deficiency [31,32]. Patients undergoing hemodialysis easily have vitamin K deficiency because of a low intake of vitamin K in food [33,34]. Therefore, patients undergoing hemodialysis in our study receiving CFP–SUL are likely to have abnormal coagulation due to vitamin K deficiency secondary to low intake of vitamin K rather than dosage of CFP–SUL.

Our study has several limitations. First, the retrospective study design has inherent limitations due to potential confounding factors and selection bias. Although it was a PSM designed study, which included sex, age, CCI, and PBS, other potential confounding factors, such as primary source of infection or the culprit of septicemia, were still not well controlled in this comparative outcome study. Second, patients with severe liver disease, which may affect PT, were not excluded from this study. However, only two patients with severe liver disease were included, one in each group. Third, the sample size of MDRO bacteremia was limited, and the majority of pathogens differed between the two groups. This may contribute to the difference in mortality among patients with bacteremia due to MDROs. Moreover, the endpoint was all-cause mortality. Mortality may not be related to septicemia. Fourth, there was no minimal inhibitory concentration (MIC) data for CFP–SUL regarding the bacteria involved in our study. Some studies suggest that high MIC values are associated with poor outcomes, especially in nonfermentative Gram-negative bacilli [35]. More pharmacokinetic–pharmacodynamic studies regarding CFP–SUL and these bacteria are needed. Lastly, there were only few patients with eGFR of <30 mL/min/1.73 m^2^ or receiving renal replacement therapy as the definition of impaired renal function in our study was an eGFR of <60 mL/min/1.73 m^2^. Further study conducted in patients with eGFR of <30 mL/min/1.73 m^2^ is necessary to confirm the results of our study.

## 4. Materials and Methods

### 4.1. Data Source

This retrospective cohort study was conducted using the Chang Gung Research Database (CGRD). CGRD is an electronic health record dataset from the healthcare system of Chang Gung Memorial Hospital, which is composed of two medical centers and five local hospitals in Taiwan [36]. Chang Gung Memorial Hospital provides approximately 10% of all healthcare services within the Taiwan National Health Insurance program, which is a single-payer nationwide health insurance program that covers over 99% of Taiwan’s population [37]. The CGRD contains variable information on detailed diagnosis, prescription, and laboratory test results from the emergency department, inpatient, and outpatient settings. The current study was approved by the Institutional Review Board of the Chang Gung Medical Foundation, Taipei, Taiwan (201901165B0C501). All personal identifiable information was anonymized; therefore, the need for informed consent was waived. All methods were performed in accordance with the relevant guidelines and regulations.

### 4.2. Study Cohort

We identified febrile inpatients from the CGRD, aged over 20 years, with any positive blood culture, and with a prescription of CFP–SUL between 1 January 2015, and 30 June 2019. Patients were excluded if they had bacteremia with Gram-positive organisms or fungi, had received CFP–SUL within 1 week before the septicemic episode, had an estimated glomerular filtration rate (eGFR) of >60 mL/min/1.73 m^2^, had received continuous renal replacement therapy, CFP–SUL course for <7 days, or received concomitant warfarin or direct oral anticoagulant. If the patients had more than one episode of bacteremia, only the first episode was included. Patients included in the study were classified into two groups: CFP–SUL 1 g/1 g and 2 g/2 g. PSM with a 1:3 ratio by sex, age, CCI [38], and PBS [39] was employed to balance the distribution of baseline characteristics between the two groups. The clinical severity at the time of blood sampling for cultures was stratified using PBS. Patients with a PBS score of ≥4 points were considered to have a critical condition [39]. The Gram-negative bacteria isolated from the included patients were susceptible to CFP–SUL. Antimicrobial susceptibility testing was determined using disk diffusion method and interpreted according to the breakpoints suggested by a previous study [40]. 

### 4.3. Outcome

The primary outcome of interest was 14-day and 30-day all-cause mortality rates. Other outcomes included clinical resolution, coagulopathy, CDAD, diarrhea, leukopenia, and neutropenia. Clinical resolution was defined as having no fever, hypothermia, leukocytosis, or leukopenia 1 day before and after the end of treatment. Coagulopathy was defined as any abnormal level of international normalized ratio (INR) within 30 days after starting to receive CFP–SUL. CDAD was defined as a patient who tested positive for stool *C. difficile* toxin gene or stool culture for *C. difficile* within 1 week of the end of treatment. Diarrhea was defined as the use of dioctahedral smectite or loperamide within 1 week of the end of treatment. Leukopenia and neutropenia were defined as white blood cell count of <4000/μL of blood and absolute neutrophil count <1500/μL of blood within 1 week of the end of treatment.

We also examined the clinical outcomes and adverse effects of patients receiving different dosages of CFP–SUL for bacteremia due to MDROs. Bacteremia due to MDROs was defined as an episode of bloodstream infection caused by ceftriaxone-resistant *Enterobacteriaceae*, *Pseudomonas* spp., or *Acinetobacter* spp.

### 4.4. Covariates

Sex, age, CCI, and PBS were identified using the ICD-9/10-CM codes from outpatient visits or hospital discharges. ICD codes for diseases composed of CCI are listed in Appendix A. PBS was graded within 48 h before or on the day of the first positive blood culture, and the highest point score during that time was recorded.

### 4.5. Statistical Analysis

Continuous variables are reported as mean, median, minimum, maximum, and standard deviation, and categorical variables are reported as numbers and percentages. The Mann–Whitney *U* test was performed to compare continuous variables, and two-sided Fisher’s exact or Pearson’s chi-square (χ2) tests were performed to compare categorical variables between the groups. A 1:3 PSM study group was created to minimize the confounding effects of the baseline characteristics. Propensity scores were calculated using a logistic regression model, including sex, age, CCI, and PBS. Univariate and multivariate logistic regression analyses were performed to determine factors associated with mortality. The Hosmer–Lemeshow goodness-of-fit test was performed to evaluate the predictive performance of the logistic regression model. A *p*-value of <0.05 was set to determine statistically significant differences.

## 5. Conclusions

Among patients with bacteremia caused by Gram-negative bacteria, we found no differences in abnormal INR, other adverse effects, or all-cause mortality between patients receiving CFP–SUL 1 g/1 g Q12H and CFP–SUL 2 g/2 g Q12H. Nevertheless, the study may have been underpowered to identify a significant difference in all-cause mortality or other clinical outcomes. More studies with pharmacokinetic–pharmacodynamic optimization are required to verify the effects of different dosages of CFP–SUL in patients with renal insufficiency.

## Figures and Tables

**Figure 1 antibiotics-11-00460-f001:**
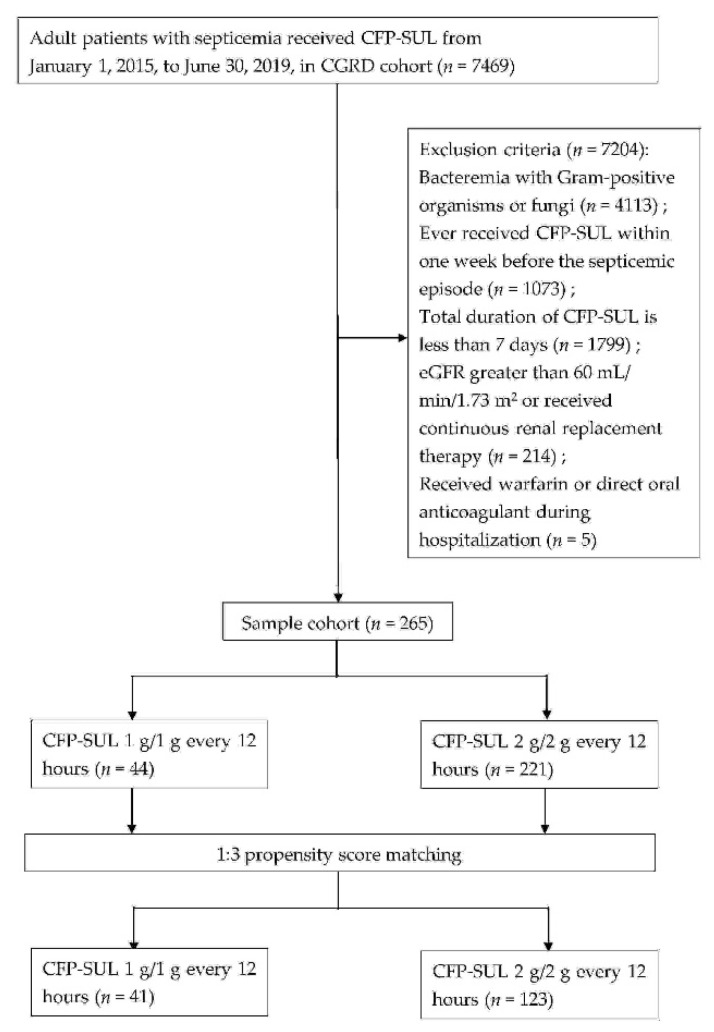
Flow chart of patient selection.

**Figure 2 antibiotics-11-00460-f002:**
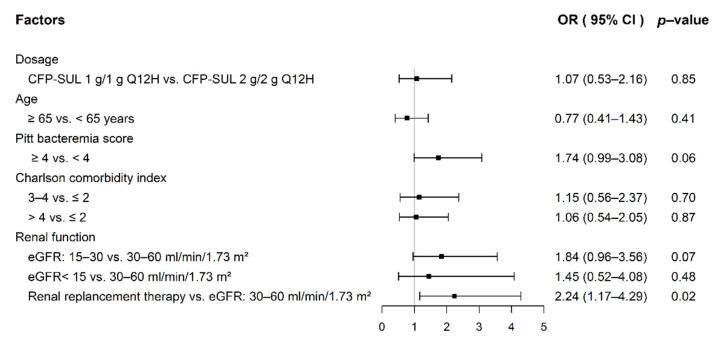
Risk factors for abnormal INR in septicemic patients with impaired renal function.

**Table 1 antibiotics-11-00460-t001:** Clinical characteristics of CGRD cohort before and after propensity score matching (PSM).

		Unmatched Cohort (*n* = 265)					
		Total	CFP–SUL 1 g/1 g Q12H (*n* = 44) (%)		CFP–SUL 2 g/2 g Q12H (*n* = 221) (%)		*p*-Value
Sex							
	Male	158	21	(47.7)	137	(62.0)	0.08
	Female	107	23	(52.3)	84	(38.0)	
Age							0.46
	Mean	74.9	76.3		74.7		
	Standard deviation	13.0	12.4		13.1		
	Median	77.7	79.3		77.0		
	Minimum	29.3	47.8		29.3		
	Maximum	102.9	96.8		102.9		
Charlson comorbidity index (CCI)							0.01
	Mean	2.6	3.6		2.4		
	Standard deviation	3.2	3.5		3.1		
	Median	1.0	3.0		1.0		
	Minimum	0.0	0.0		0.0		
	Maximum	16.0	12.0		16.0		
Pitt bacteremia score (PBS)							0.86
	Mean	2.1	2.2		2.1		
	Standard deviation	2.5	2.6		2.5		
	Median	1.0	1.0		1.0		
	Minimum	0.0	0.0		0.0		
	Maximum	10.0	9.0		10.0		
Body weight							0.48
	Mean	54.4	53.7		54.5		
	Standard deviation	6.6	6.8		6.5		
	Median	55.0	53.0		55.7		
	Minimum	30.0	40.0		30.0		
	Maximum	74.0	69.0		74.0		
Comorbidity							
	Myocardial infarction	12	1	(2.3)	11	(5.0)	0.43
	Congestive heart failure	29	9	(20.5)	20	(9.0)	0.03
	Peripheral vascular disease	4	1	(2.3)	3	(1.4)	0.65
	Cerebrovascular disease	46	8	(18.2)	38	(17.2)	0.87
	Dementia	3	1	(2.3)	2	(0.9)	0.43
	Chronic pulmonary disease	39	7	(15.9)	32	(14.5)	0.81
	Connective tissue disease	3	0	(0.0)	3	(1.4)	0.44
	Peptic ulcer disease	28	4	(9.1)	24	(10.9)	0.73
	Mild liver disease	19	8	(18.2)	11	(5.0)	0.00
	Diabetes without end-organ damage	52	11	(25.0)	41	(18.6)	0.33
	Diabetes with end-organ damage	25	4	(9.1)	21	(9.5)	0.93
	Hemiplegia	4	0	(0.0)	4	(1.8)	0.37
	Moderate or severe renal disease	55	13	(29.5)	42	(19.0)	0.12
	Tumor without metastasis (include leukemia and lymphoma)	47	11	(25.0)	36	(16.3)	0.17
	Severe liver disease	2	1	(2.3)	1	(0.5)	0.20
	Metastatic solid tumor	30	8	(18.2)	22	(10.0)	0.12
Renal function							
	eGFR: 30–60	115	14	(31.8)	101	(45.7)	0.35
	eGFR: 15–30	62	11	(25.0)	51	(23.1)	
	eGFR: <15	19	4	(9.1)	15	(6.8)	
	Renal replacement therapy	69	15	(34.1)	54	(24.4)	
		**PSM Cohort (*n* = 164)**					
		**Total**	**CFP–SUL 1 g/1 g Q12H (*n* = 41) (%)**		**CFP–SUL 2 g/2 g** **Q12H (*n* = 123) (%)**		***p*-Value**
Sex							
	Male	83	21	(51.2)	62	(50.4)	0.93
	Female	81	20	(48.8)	61	(49.6)	
Age							0.54
	Mean	75.8	76.5		75.5		
	Standard deviation	11.9	12.4		11.7		
	Median	77.2	79.3		76.1		
	Minimum	47.8	47.8		52.7		
	Maximum	102.9	96.8		102.9		
Charlson comorbidity index (CCI)							0.95
	Mean	3.1	3.1		3.1		
	Standard deviation	3.2	3.1		3.2		
	Median	2.0	3.0		2.0		
	Minimum	0.0	0.0		0.0		
	Maximum	12.0	12.0		12.0		
Pitt bacteremia score (PBS)							0.90
	Mean	2.2	2.1		2.2		
	Standard deviation	2.5	2.5		2.5		
	Median	1.0	1.0		1.0		
	Minimum	0.0	0.0		0.0		
	Maximum	9.0	9.0		8.0		
Body weight							0.78
	Mean	53.4	53.7		53.3		
	Standard deviation	6.7	7.1		6.6		
	Median	52.8	52.8		53.0		
	Minimum	34.0	40.0		34.0		
	Maximum	74.0	69.0		74.0		
Comorbidity							
	Myocardial infarction	10	1	(2.4)	9	(7.3)	0.26
	Congestive heart failure	23	8	(19.5)	15	(12.2)	0.24
	Peripheral vascular disease	3	1	(2.4)	2	(1.6)	0.74
	Cerebrovascular disease	32	8	(19.5)	24	(19.5)	>0.99
	Dementia	1	1	(2.4)	0	(0.0)	0.08
	Chronic pulmonary disease	26	7	(17.1)	19	(15.4)	0.81
	Connective tissue disease	2	0	(0.0)	2	(1.6)	0.41
	Peptic ulcer disease	18	3	(7.3)	15	(12.2)	0.39
	Mild liver disease	13	6	(14.6)	7	(5.7)	0.07
	Diabetes without end-organ damage	37	9	(22.0)	28	(22.8)	0.91
	Diabetes with end-organ damage	20	3	(7.3)	17	(13.8)	0.27
	Hemiplegia	2	0	(0.0)	2	(1.6)	0.41
	Moderate or severe renal disease	41	11	(26.8)	30	(24.4)	0.75
	Tumor without metastasis (include leukemia and lymphoma)	35	9	(22.0)	26	(21.1)	0.91
	Severe liver disease	1	0	(0.0)	1	(0.8)	0.56
	Metastatic solid tumor	24	6	(14.6)	18	(14.6)	>0.99
Renal function							
	eGFR: 30–60	68	13	(31.7)	55	(44.7)	0.52
	eGFR: 15–30	40	11	(26.8)	29	(23.6)	
	eGFR: <15	10	3	(7.3)	7	(5.7)	
	Renal replacement therapy	46	14	(34.1)	32	(26.0)	
		**Unmatched Cohort (*n* = 265)**			**PSM Cohort (*n* = 164)**		
		**CFP–SUL 1 g/1 g Q12H (*n* = 44)**	**CFP–SUL 2 g/2 g Q12H (*n* = 221)**	**SMD ***	**CFP–SUL 1 g/1 g Q12H (*n* = 41)**	**CFP–SUL 2 g/2 g Q12H (*n* = 123)**	**SMD ***
Male (%)		21 (47.7)	137 (62.0)	0.3	21 (51.2)	62 (50.4)	0.0
Age (SD)		76.3 (12.4)	74.7 (13.1)	0.1	76.5 (12.4)	75.5 (11.7)	0.1
CCI (SD)		3.6 (3.5)	2.4 (3.1)	0.4	3.1 (3.1)	3.1 (3.2)	0.0
PBS (SD)		2.2 (2.6)	2.1 (2.6)	0.0	2.2 (2.5)	2.2 (2.5)	0.0

SMD *: standardized mean differences; SMD < 0.1 was considered well balanced. CGRD, Chang Gung Research Database; PSM, propensity score matching; CFP–SUL, cefoperazone–sulbactam; eGFR, estimated glomerular filtration rate; CCI, Charlson comorbidity index.

**Table 2 antibiotics-11-00460-t002:** Clinical outcomes and adverse effects of unmatched and matched CGRD cohort.

	Unmatched Cohort (*n* = 265)					PSM Cohort (*n* = 164)				
	CFP–SUL 1 g/1 g Q12H (*n* = 44)		CFP–SUL 2 g/2 g Q12H (*n* = 221)		*p*-Value	CFP–SUL 1 g/1 g Q12H (*n* = 41)		CFP–SUL 2 g/2 g Q12H (*n* = 123)		*p*-Value
	n	(%)	n	(%)		n	(%)	N	(%)	
14-day all-cause mortality	11	(25.0)	47	(21.3)	0.58	10	(24.4)	28	(22.8)	0.83
30-day all-cause mortality	14	(31.8)	55	(24.9)	0.34	13	(31.7)	32	(26.0)	0.48
Clinical resolution	32	(72.7)	162	(73.3)	0.94	29	(70.7)	91	(74.0)	0.68
Abnormal INR	17	(38.6)	83	(37.6)	0.89	15	(36.6)	50	(40.7)	0.64
*Clostridioides difficile*- associated diarrhea	1	(2.3)	4	(1.8)	0.84	1	(2.4)	0	(0.0)	0.25
Diarrhea	8	(18.2)	67	(30.3)	0.10	8	(19.5)	33	(26.8)	0.35
Leukopenia	2	(4.5)	14	(6.3)	0.65	2	(4.9)	6	(4.9)	>0.99
Neutropenia	1	(2.3)	9	(4.1)	0.57	1	(2.4)	6	(4.9)	0.68

CGRD, Chang Gung Research Database; PSM, propensity score matching; INR, international normalized ratio.

**Table 3 antibiotics-11-00460-t003:** (a) Clinical outcomes and adverse effects of patients receiving CFP–SUL for septicemia caused by MDRO, (b) Mortality rate of septicemia caused by MDRO: subgroup analysis.

(a)					
	CFP–SUL 1 g/1 g Q12H (*n* = 23)		CFP–SUL 2 g/2 g Q12H (*n* = 102)		*p*-Value
	n	(%)	n	(%)	
14-day all-cause mortality	2	(8.7)	21	(20.6)	0.24
30-day all-cause mortality	3	(13.0)	26	(25.5)	0.20
Clinical resolution	18	(78.3)	80	(78.4)	>0.99
Abnormal INR	7	(30.4)	38	(37.3)	0.54
*Clostridioides difficile*-associated diarrhea	1	(4.3)	2	(2.0)	0.46
Diarrhea	3	(13.0)	31	(30.4)	0.09
Leukopenia	1	(4.3)	9	(8.8)	0.69
Neutropenia	1	(4.3)	5	(4.9)	>0.99
**(b)**					
	**CFP–SUL 1 g/1 g Q12H** **(*n* = 23)**		**CFP–SUL 2 g/2 g Q12H** (***n* = 102)**		***p*-Value**
	**n**	**(%)**	**n**	**(%)**	
Number of patients receiving CFP–SUL					
*Pseudomonas* spp.	4	(17.4)	19	(18.6)	<0.01
*Acinetobacter* spp.	1	(4.3)	40	(39.2)	
Ceftriaxone-resistant *Enterobacteriaceae*	18	(78.3)	43	(42.2)	
14-day all-cause mortality					
*Pseudomonas* spp.	0	(0.0)	3	(15.8)	>0.99
*Acinetobacter* spp.	0	(0.0)	11	(27.5)	>0.99
Ceftriaxone-resistant *Enterobacteriaceae*	2	(11.1)	7	(16.3)	0.71
30-day all-cause mortality					
*Pseudomonas* spp.	1	(25.0)	4	(21.1)	>0.99
*Acinetobacter* spp.	0	(0.0)	14	(35.0)	>0.99
Ceftriaxone-resistant *Enterobacteriaceae*	2	(11.1)	8	(18.6)	0.47

MDRO, multidrug-resistant organisms.

**Table 4 antibiotics-11-00460-t004:** Univariable and multivariable logistic regression analysis of factors in CGRD cohort associated with 14-day or 30-day all-cause mortality.

		Univariable		Multivariable		Hosmer–Lemeshow Goodness-of-Fit Test
Factors	Comparisons	OR (95% CI)	*p*-Value	OR (95% CI)	*p*-Value	
14-day all-cause mortality						14.93 (*p* = 0.06)
CCI	Per 1-score increase	1.07 (0.98–1.16)	0.14	1.10 (1.00–1.20)	0.05	
PBS	≥4 vs. <4	1.78 (0.96–3.32)	0.07	2.07 (1.08–3.96)	0.03	
Dosage	1 g/1 g vs. 2 g/2 g	1.23 (0.58–2.62)	0.59	1.11 (0.50–2.45)	0.80	
Sex	Male vs. female	1.38 (0.75–2.53)	0.30	1.36 (0.72–2.55)	0.35	
Age	Per 1-year increase	1.03 (1.01–1.06)	0.02	1.04 (1.01–1.06)	0.01	
30-day all-cause mortality						5.92 (*p* = 0.66)
CCI	Per 1-score increase	1.04 (0.96–1.13)	0.32	1.06 (0.97–1.15)	0.19	
PBS	≥4 vs. <4	1.85 (1.03–3.35)	0.04	2.01 (1.09–3.69)	0.03	
Dosage	1 g/1 g vs. 2 g/2 g	1.41 (0.70–2.85)	0.34	1.34 (0.65–2.80)	0.43	
Sex	Male vs. female	1.38 (0.78–2.44)	0.27	1.38 (0.77–2.49)	0.28	
Age	Per 1-year increase	1.02 (1.00–1.04)	0.12	1.02 (1.00–1.04)	0.09	

CCI, Charlson comorbidity index; PBS, Pitt bacteremia score.

## Data Availability

The datasets generated and/or analyzed during the current study are available from the corresponding author upon reasonable request.

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
