# Peer review of "Clinical Outcomes and Adverse Effects in Septic Patients with Impaired Renal Function Who Received Different Dosages of Cefoperazone–Sulbactam"

_antibiotics, 2022, doi:10.3390/antibiotics11040460_

Round 1

Reviewer 1 Report

This retrospective study aimed to compare clinical outcomes in septic patients caused by Gram-negative bacteria with impaired renal function who received different dosages of cefoperazone-sulbactam.   In conclusion, there were no significant between-group differences in all-cause mortality rates and adverse effects. This study is very important, and the cohort of study is sufficient. Moreover, there are important errors of calculation in my opinion, and the revision of the study should be done with very attention by the Authors.

1) Line 34: add this reference: PMID: 34356772 and comment it; Line 46: add this reference: PMID: 33260698 and comment it.

2) Line 49: …combined with additional antibacterial agents… Which are these drugs, improve this concept.

3) Line 69: move figure 1 to this line, and improve the figure.

4) Line 19,70: Gram-positive, check in the text and correct it.

5) Table 3-1:  “Clinical resolution= 80, 14-day all-cause mortality=21 and 30-day all-cause mortality= 26, but the total is n=102…but in the first group, the count is true. Correct and check in the text.

6) Line: 136-139: In table 3.1, the data reported a less of mortality rate in group 1 than in group 2, but in these lines, you refer to different results. It is very confusing.

7) The Discussion section, format the text character, justifiable.

8) Carefully check the calculations and data in the tables and text.

Reviewer 2 Report

Your papes is interesting to have an alternative drug in the treatment of infection caused by Acinetobacter baumannii or Pseudomonas aeruginosa in patients with impaired renal function

As you say at the end of the paper "More studies are required to verify her effects of diffent dosages of CFP-SUL...".  As you know, one of the first thing to study when  analyse the effect of an antibiotic or a combination of Beta-lactam and Beta-lactamase inhibitor  is the MIC of the antibiotic versus bacteria involved in infection. In this paper there aren't any MIC to understand if the concentration of antibiotics utilised  are effective against the microorganisms considered. I know that this is not the aim of your work, but is one of ther first thing to know in interaction antibiotic-bacteria. You only say  (lines 53-56) that "a pk-pd study suggests a higher dosage of sulbactam (2g every 6h) for patients with renal insufficiency in the treatment of MDR-AB infection may be utilised". 

Lines 117-119 "The proportion of infection caused by Acinetobacter spp. was significantly higher in patients treated with CFP-SUL 2g/2g group than in CFP-SUL 1g/1g".  Therefore, CFP-SUL 2g/2g is more effective than CFP-SUL 1g/1g, or not?

Moreover,  the utilisation of sulbactam in high dose for long time, induce the production of other Beta-lactamase, such as AmpC. This particularity must be considered.

Reviewer 3 Report

The manuscript by Tai et al, “Clinical Outcomes and Adverse Effects in Septic Patients with Impaired Renal Function Who Received Different Dosages of Cefoperazone-Sulbactam”. In this study authors aimed to compare clinical outcomes and adverse effects in septic patients with an impaired renal function who received different dosages of cefoperazone-sulbactam. A total of 265 patients were divided into two groups were included in this study. The authors did not find any significant difference between-group differences in all-cause mortality rates and adverse effects, including prolonged prothrombin time. After the logistic regression model, they have shown that the Pitt bacteremia score was related to all-cause mortality rate, and the prolonged prothrombin time was associated with renal replacement therapy.

The overall study was conducted very carefully, and the introduction, methods, results and discussion were sufficiently described and well presented. I do feel this manuscript is suitable for the publication in Antibiotic journal. However, I suggest authors improve their English language or use native English writers to improve language of the manuscript.

Round 2

Reviewer 1 Report

The article has been improved and the suggestions have been verified. Congratulations for the excellent work, I invite you to publish the manuscript.

Author Response

Dear reviewer, 

Thank you for your suggestions and instructions of this article. 

We are grateful for these opinions. 

Yours sincerely, 

Chien-Hsiang Tai